# Learning to Learn with Feedback and Local Plasticity

**Jack Lindsey**
Columbia University, Department of Neuroscience
j.lindsey@columbia.edu

## Abstract

Developing effective biologically plausible learning rules for deep neural networks is important for advancing connections between deep learning and neuroscience. To date, local synaptic learning rules like those employed by the brain have failed to match the performance of backpropagation in deep networks. In this work, we employ meta-learning to discover networks that learn using feedback connections and local, biologically motivated learning rules. Importantly, the feedback connections are not tied to the feedforward weights, avoiding any biologically implausible weight transport. It can be shown mathematically that this approach has sufficient expressivity to approximate any online learning algorithm. Our experiments show that the meta-trained networks effectively use feedback connections to perform online credit assignment in multi-layer architectures. Moreover, we demonstrate empirically that this model outperforms a state-of-the-art gradient-based meta-learning algorithm for continual learning on regression and few-shot classification benchmarks. This approach represents a step toward biologically plausible learning mechanisms that can not only match gradient descent-based learning, but also overcome its limitations.

## 1   Introduction

Deep learning has achieved impressive success in solving complex tasks, and in some cases its learned representations have been shown to match those in the brain [19, 10]. However, there is much debate over how well the learning algorithm commonly used in deep learning, backpropagation, resembles biological learning algorithms. Causes for skepticism include the facts that (1) backpropagation ignores the nonlinearities imposed by neurons in the backward pass and assumes instead that derivatives of the forward-pass nonlinearities can be applied, (2) in backpropagation, feedback path weights are exactly tied to feedforward weights, even as weights are updated with learning, and (3) backpropagation assumes alternating forward and backward passes [12]. The question of how so-called credit assignment – appropriate propagation of learning signals to non-output neurons – can be performed in biologically plausible fashion in deep neural networks remains open.

We propose a new learning paradigm that aims to solve the credit assignment problem in more biologically plausible fashion. Our approach is as follows: (1) endow a deep neural network with feedback connections that propagate information about target outputs to neurons at all layers, (2) apply local plasticity rules (e.g. Hebbian or neuromodulated plasticity) to update feedforward synaptic weights following feedback projections, and (3) employ meta-learning to optimize for the initialization of feedforward weights, the setting of feedback weights, and synaptic plasticity levels. On a set of online regression and classification learning tasks, we find that meta-learned deep networks can successfully perform useful weight updates in early layers, and that feedback with local learning rules can in fact outperform gradient descent as an inner-loop learning algorithm on challenging few-shot and continual learning tasks.

## 2   Related Work

Some research has investigated alternative algorithms to backpropagation that relax or eliminate the requirement of weight symmetry. A surprising set of results [14, 17], show that random feedback

weights are sufficient to induce learning for simple tasks. Another family of methods, known as target propagation, use a reconstruction loss to learn a feedback pathway that approximates the inverse of the feedforward pathway [3]. However, both of these approaches have been found not to scale well to difficult tasks such as ImageNet classification [2]. To some extent, performance can be recovered by permitting sign-symmetry in forward and backward weights [13], but this partially re-introduces the weight symmetry issue and fails to address concerns (1) and (3) above.

Backpropagation-based deep learning notably falls short of human and animal learning in several key respects. In particular, it has difficulty learning from few examples, and learning in online fashion from a stream of data and on multiple tasks. One approach to addressing these issues is meta-learning, in which a network's learning procedure itself is learned in an "outer loop" of optimization. A popular class of such methods is gradient-based meta-learning [4], in which the network initialization is meta-optimized so that batch gradient descent will learn quickly from few examples of a new task. In the batch (i.e. not online) learning case, this approach has the expressive power to implement any batch learning algorithm [5]. This method has been extended to the continual learning case, in which the "inner loop" optimization consists of many online gradient steps on a potentially nonstationary data distribution [8].

Building on the meta-learning paradigm, another line of research has explored the approach of performing inner-loop updates according to biologically motivated Hebbian learning rules rather than by gradient descent [1, 15, 16]. However, none of these methods fully address the credit assignment problem, in that they either restrict plasticity to output weights or allow plasticity to proceed without any dependence on supervised error signals. Recent work has also considered meta-learning algorithm for learning feedback weights [11]. Their methods, based on node perturbation and RL algorithms, differ substantially from ours, but a comparison or synthesis could prove fruitful.

## 3   Method

See Figure 1 for a schematic comparing our model to standard backpropagation and direct feedback alignment [17]. In our model, a network propagates an input $\mathbf{x}$ forward through a neural network $\hat{f}(\cdot; \theta)$, receives a target signal $\mathbf{y}$ from the environment, and propagates a function $g(\mathbf{y})$ back to its neurons. The output of $g$ is an update to the activations of the network. Subsequently, the network undergoes synaptic plasticity according to a local learning rule that adjusts a synaptic weight $w$ based on the previous weight value, the presynaptic activity $a$, and the postsynaptic activity $b$ resulting from feedback. In some experiments we allow plasticity only in the final $N$ network layers (varying $N$).

We may take $a$ to be the pre or post-feedback presynaptic activations. The post-feedback case corresponds to a model in which neural activations are updated directly with feedback and Hebbian-style plasticity ensues. The pre-feedback case requires error signals to be propagated without affecting the neural activations used in feedforward computation. This approach has the advantage of avoiding possible disruption to the feedforward computation, though it may be more difficult to implement. Possible biological implementations include a segregated dendrites model (see [6]), or feedback through neuromodulatory signals at postsynaptic sites, with weight updates that are proportional to presynaptic and neuromodulatory activity, but not postsynaptic activity (see [7]). In our simulations we use Oja's learning rule: $w \leftarrow w + \alpha(ab - b^2 w)$, wher $\alpha$ is a plasticity coefficient [18].

We use linear feedback connections with one ReLU nonlinearity applied to enforce positive-valued feedback activations. Concretely, the activation $x_i$ at the output of layer $W_i$ with corresponding feedback matrix $G_i$ is set to $\text{ReLU}[(1 - \beta_i)x_i + \beta_i G_i \mathbf{y}]$, where $\beta_i$ controls the "strength" of feedback. Note that $\beta_i = 0$ corresponds to pure unsupervised Hebbian learning in layer $i$, while $\beta_i = 1$ corresponds to supervised learning.

### 3.1   Meta-learning procedure

The description above specifies how a network in our model learns in its "lifetime." However, to create a network that effectively learns using the above procedure, we employ meta-learning. In particular, for each of our benchmark tasks (described below) we simulate an entire learning episode and test input, evaluate the performance on the test input, and backpropagate through the entire learning procedure (see [4, 8]). The meta-learned parameters are the initial weights $\theta$ and feedback

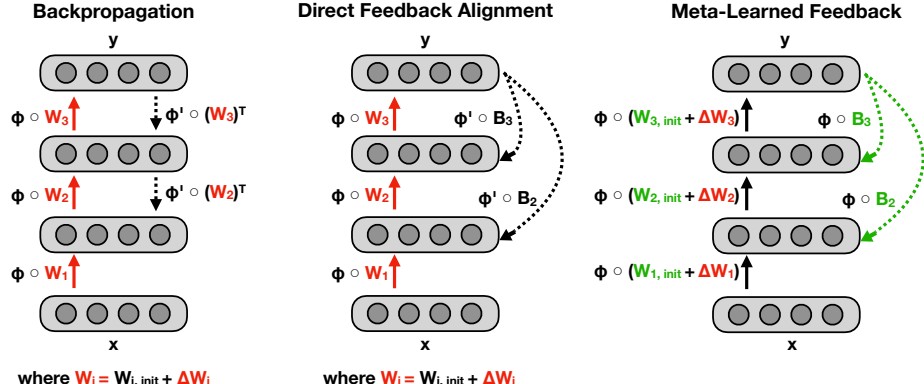

Figure 1: A comparison of standard backpropagation, direct feedback alignment [17], and the proposed method. W variables represent linear transformations, indicates a neuron's activation function, and denotes composition. Red quantities indicate plastic weights that change during a network's lifetime, while green quantities indicate meta-learned quantities optimized over many lifetimes. In backpropagation, learning signals propagate through a feedback pathway involving transposes of the feedforward weights and the derivative of the neuron activation function. Direct feedback alignment replaces the transpose matrices with random feedback pathways. In the proposed method, feedforward weights evolve according to Hebbian plasticity during a lifetime, while feedback pathways and initial feedforward weights are meta-optimized over lifetimes. Additionally, error signals are injected into layers directly, without any derivative computations.

function $g$, as well as the plasticity coefficients for each plastic weight and each layer's $\beta$ coefficient, which controls the balance of supervised vs. unsupervised learning occurring in that layer.

## 3.2 Universiality

We are able prove that sufficiently wide and deep neural networks using the above learning procedure can approximate any learning algorithm. A learning algorithm, for our purposes, maps a set of training examples $\{(\mathbf{x}, \mathbf{y})_k\}$ and a test input $\mathbf{x}^\star$ to a predicted output $\hat{\mathbf{y}}^*$.

**Theorem.** For any learning rule $f_{\text{target}}(\{(\mathbf{x}, \mathbf{y})_k\}, \mathbf{x}^\star)$, there exists a deep ReLU network feedforward function $\hat{f}(\cdot; \theta)$ and feedback function $g(\mathbf{y})$ such that $\hat{f}(\mathbf{x}^\star; \theta') \approx f_{\text{target}}(\{(\mathbf{x}, \mathbf{y})_k\}, \mathbf{x}^\star)$. Here $\theta' = \theta_k$, $\theta_0 = \theta$, and $\theta_{k+1} = \theta_k + \Delta_{\theta_k}(\mathbf{y}, \mathbf{x})$, where $\Delta_\theta(\mathbf{y}, \mathbf{x})$ is computed following feedback according to a local learning rule at each synapse, either Hebb's rule or Oja's rule.

**Proof.** See Appendix A. It borrows techniques from [5], which proved a similar universality result for gradient-based meta-learning in the non-online batch learning case (in which the entire dataset is available at once). The feedforward network initialization and feedback weights are chosen so that the weight updates losslessly encode the training data in early layers of the network in such a way that it can be processed in an arbitrary way (i.e. to simulate $f_{\text{target}}$) by downstream layers. The proof deviates from [5] in at least one major respect: in the online, continual case, the ability to choose the feedback weights is essential. Indeed, one can indeed show that there are some reasonable $f_{\text{target}}$ which gradient-based learning (where feedback weights are tied to feedforward weights) cannot approximate.

## 4 Experiments

We build off the experimental protocol of [8], evaluating our approach on a regression task and a classification task, all requiring online continual learning. We use the same architectures for the regression and classification tasks as [8] (in short, a nine-layer fully connected network for regression and an six convolutional layer + two fully connected layer for classification).

**Incremental Sine Waves:** The regression problem is as follows: in each training episode, ten sine functions are sampled randomly, parameterized by amplitude in $[0.1, 5]$ and phase in $[0, \pi]$. In each episode, forty size-32 batches of $(\tilde{x}, y)$ pairs from the first function are presented, then forty from the

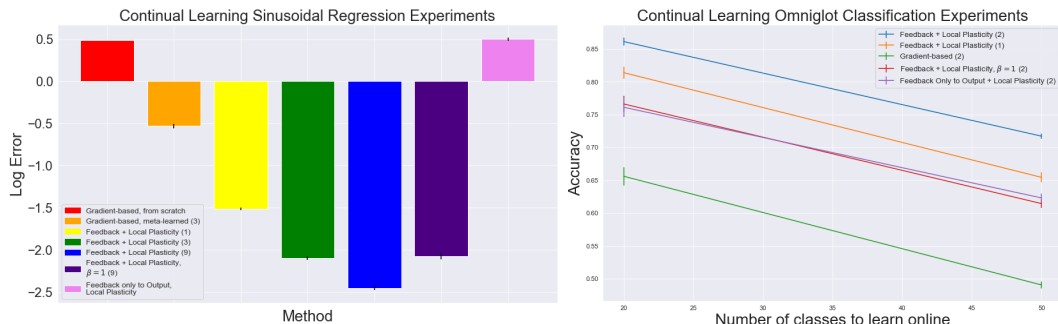

Figure 2: Performance of models on regression and one-shot classification experiments. For regression we show performance after learning all 40 classes. In both cases, meta-learned networks which learn with feedback and local plasticity outperform gradient-based meta-learners. Ablation experiments with local plasticity alone show the necessity of feedback. Error bars indicate standard error over 50 evaluations of the trained network. Numbers in parentheses show the number of plastic layers, starting with the output.

second, and so on. The input $\tilde{x}$ contains both the function input $x$ and the index $k$ of the function being used. The network is tasked with outputing $y$ for a new $\tilde{x}$. Evaluation occurs on new episodes with sine functions not used in meta-training. Meta-training is performed for 20,000 episodes.

**Split-Omniglot:** The dataset is split into meta-training and meta-testing classes. During an episode, $k$ examples from one class are presented, followed by $k$ from the next, up to a total of $N$ classes. The model is tested on unseen examples from the classes in the episode. We choose $k = 1$, $N = 20$ to consider the one-shot continual learning case. Feedback to output activations is clamped to their target values, but feedback weights to earlier layers are meta-learned. Evaluation occurs on episodes with classes not used in meta-training. Meta-training is performed for 40,000 episodes.

**Experimental Protocol:** We evaluate our method in a number of ways: (1) We compare its performance to a gradient-based meta-learner with the same architecture (we also allow its plasticity coefficients to be meta-learned to permit fair comparison). (2) We vary the number of plastic layers in the network. In particular, the case in which only the output weights are plastic serves as a control to indicate whether the feedback propagation to earlier layers is indeed helping learning. (3) We perform ablation experiments to discern the significance of the learned feedback weights. In particular, we experiment with disallowing feedback altogether but maintaining Hebbian plasticity throughout the network, and with clamping all $\beta$ parameters to 1 to prevent the network from performing Hebbian unsupervised learning along with feedback-modulated updates. (4) We compare using pre or post-feedback presynaptic activations for plasticity updates, corresponding to the two scenarios (with or without dendritically segragated or neuromodulator-carried learning signals) described above.

## 5 Results

Experimental outcomes are shown in Figure 2. We find that the architecture with meta-learned feedback and local plasticity significantly outperforms an architecturally equivalent gradient-based meta-learner on both the regression and classification tasks. Ablation experiments show that feedback in addition to local plasticity is necessary to enable learning, and that feedback to earlier layers aids performance beyond what can be achieved with feedback only to output layers. Interestingly, we also find that networks invariably learn $\beta$ values between 0 and 1, and that networks with all $\beta$ fixed at 1 perform worse. This result indicates that a mix of unsupervised Hebbian and supervised feedback-modulated learning is beneficial. We additionally examined the correlation between weight updates in the feedback network and updates that *would* be computed by gradient descent. We find that the average correlation between the two increases from early to late layers but remains weak ($< 0.1$) throughout, and is even negative at some stages in the learning process on the regression task. This phenomenon suggests that the meta-learned feedback network learns in a manner that is qualitatively different from gradient-based learners.

# 6 Discussion

This work demonstrates that meta-learning procedures can optimize for neural networks that learn online using local plasticity rules and feedback connections. Several follow-up directions could be pursued. First, meta-learning of this kind is computationally expensive, as the meta-learner must backpropagate through the network's entire training procedure. In order to scale this approach, it will be important to find ways to meta-train networks that generalize to longer lifetimes than were used during meta-training, or to explore alternatives to backprop-based meta-training (e.g. evolutionary algorithms). The present work focused on the case of online learning, but the case of learning from repeated exposure to large datasets is also of interest, and scaling the method in this fashion will be crucial to exploring this regime.

Future work could also increase the biological plausibility of the method. For instance, in the present implementation the feedforward and feedback + update passes occur sequentially. However, a natural extension would enable them to run in parallel. This requires ensuring (through appropriate meta-learning and/or a segregated dendrites model [6]) that feedforward and feedback information do not interfere destructively. Third, the meta-learning procedure in this work optimizes for a precise feedforward and feedback weight initialization. Optimizing instead for a distribution of weight initializations or connectivity patterns would better reflect the stochasticity in synapse development. Another direction is to apply meta-learning to understand biological learning systems (see [9] for an example of such an effort). Well-constrained biological learning models meta-optimized in this manner might show emergence of learning circuits used in biology and even suggest new ones.

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

## A   Appendix: Universality Proofs

We prove that sufficiently wide and deep neural networks with supervised feedback and local learning rules can approximate any learning algorithm. We borrow some of the notation and proof techniques from [5]. We suppose the network propagates an input $\mathbf{x}$ forward, receives a target signal $\mathbf{y}$ from a supervisor, propagates a function of $\mathbf{y}$ back to its neural activations (feedback), and undergoes synaptic plasticity according to a local learning role dependent on these activations. We let $\{(\mathbf{x}_k, \mathbf{y}_k)\}$ denote the training data, observed in that order, and $\mathbf{x}^\star$ denote the test input.

We want to construct a network architecture with feedforward function $\hat{f}(\cdot; \theta)$ and feedback function $g(\mathbf{y})$ such that $\hat{f}(\mathbf{x}^\star; \theta') \approx f_{\text{target}}(\{(\mathbf{x}, \mathbf{y})_k\}, \mathbf{x}^\star)$, where $\theta' = \theta_k$, $\theta_0 = \theta$, and $\theta_{k+1} = \theta_k + \Delta_{\theta_k}(\mathbf{y}, \hat{f}(\mathbf{x}; \theta_k))$. The update $\Delta_\theta(\mathbf{y}, \hat{f}(\mathbf{x}; \theta))$ is assumed to proceed according to a local learning rule that adjust a synaptic weight $w$ based on the previous weight value, the presynaptic activity $a$, and the postsynaptic activity $b$, where the values of $a$ and $b$ are taken following feedback propagation. We will consider Hebb's learning rule: $w \leftarrow w + \alpha(ab)$ and Oja's learning rule: $w \leftarrow w + \alpha(ab - b^2 w)$.

We let $\hat{f}$ be a deep neural network with $2N + 2$ layers and ReLU nonlinearities. We will ensure nonnegativity of the activations of the intermediate $2N$ layers, allowing us to treat them as linear. This simplification allows us to write the model as follows:

$$\hat{f}(\cdot; \theta) = f_{\text{out}}\left(\left(\prod_{i=1}^{N} V_i W_i\right) \phi(\cdot; \theta_{\text{ft}}); \theta_{\text{out}}\right),$$

where $\phi(\cdot; \theta_{\text{ft}})$ is an initial neural network with parameters $\theta_{\text{ft}}$. $\prod_{i=1}^{N} W_i^2 W_i^1$ is a product of $2N$ square linear weight matrices, and $f_{\text{out}}(\cdot; \theta_{\text{out}})$ is an output neural network with parameters $\theta_{\text{out}}$. We adopt corresponding notation of $G_i^1, G_i^2$ – feedback matrices projecting a function $\varphi(\mathbf{y})$ of the target (computed with a one-layer feedback network) to the outputs of the layers $W_i^1, W_i^2$ respectively, as well as $\beta_i^1, \beta_i^2$ (feedback strength) and $\alpha_i^1, \alpha_i^2$ (plasticity coefficients at $W_i^1$ and $W_i^2$). Concretely, the activation $x_i^j$ at the output of layer $W_i^j$ is set to $\text{ReLU}((1 - \beta_i^j)x_i^j + \beta_i^j G_i^j \varphi(\mathbf{y}))$, where $\beta_i^j \in [0, 1]$. We will ensure nonnegativity of the projection so that we may ignore the ReLU. The weights of layer $W_i^j$ are then updated according to one of the following rules:

$$W_i^j \leftarrow W_i^j + \alpha_i^j x_i^j (\tilde{x}_i^j)^T \qquad \text{(Hebb's rule)}$$

$$W_i^j \leftarrow W_i^j + \alpha_i^j [x_i^j (\tilde{x}_i^j)^T - \text{diag}(x_i^j)^2 W_i^j] \qquad \text{(Oja's rule)},$$

where $\tilde{x}_i^j$ refers to the activations at the layer preceding layer $x_i^j$, and $\text{diag}(x)$ denotes a square diagonal matrix with $x$ along the diagonal. We will conduct the proofs for Hebb's rule and Oja's rule in parallel, using  as an indicator variable – a value of 1 indicates we are using Oja's rule, and 0 corresponds to Hebb's rule. Hence we may write the learning rule compactly as follows:

$$W_i^j \leftarrow W_i^j + \alpha_i^j [x_i^j (\tilde{x}_i^j)^T - \mathcal{L} \cdot \text{diag}(x_i^j)^2 W_i^j].$$

We set all $W_i^2$ to be identity matrices, all $\beta_i^2$ to 0 (rendering the values of $G_i^2$ irrelevant), all $\beta_i^1$ to 1, all $\alpha_i^2$ to be 0, and all $\alpha_i^1$ to be a constant $\alpha$ (assumed in the rest of the proof to be sufficiently small). These choices specify an architecture consisting of feedforward layers coming in groups of two. The first layer in each group consists of a general feedforward matrix $W_i^1$, which we will henceforth write simply as $W_i$. The matrix $W_i$ will undergo plasticity at rate $\alpha$ induced by the feedforward activations at its input and feedback-induced activations at its output from feedback matrix $G_i^1$ (which we will now write simply as $G_i$). The second layer is a nonplastic identity transformation which effectively "shields" $W_{i-1}$ from undergoing plasticity induced by the feedback projection $G_i$. We assume no feedback propagation to and no plasticity in the feature extractor $\phi$ or output network $f_{\text{out}}$. Thus feedforward propagation is affected only by the $W_i$, and plasticity updates following feedback propagation will only modify the $W_i$ matrices.

Now we expand $\hat{f}(\mathbf{x}^\star; \theta')$. We let $\mathbf{z}_k = \left( \prod_{i=1}^N W_i \right) \phi(\mathbf{x}_k)$. After one step, each $W_i$ is updated as follows:

$$\Delta_{W_i} = \alpha G_i \varphi(\mathbf{y}_1)\phi(\mathbf{x}_1)^T \left( \prod_{j=i+1}^N W_j \right)^T - \alpha \mathcal{L} \cdot \text{diag}(G_i\varphi(\mathbf{y}_1))^2 W_i.$$

and up to terms of $O(\alpha^2)$, the update is of the same form for all steps $k = 1, 2, ..., K$. We let $\alpha$ be small enough that higher-order terms in $\alpha$ can be ignored. Now

$$\Delta_{W_i} = \sum_{k=1}^K \left[ \alpha G_i \varphi(\mathbf{y}_k)\phi(\mathbf{x}_k)^T \left( \prod_{j=i+1}^N W_j \right)^T - \alpha \mathcal{L} \cdot \text{diag}(G_i\varphi(\mathbf{y}_k))^2 W_i \right] + O(\alpha^2).$$

Thus we can expand $\prod_{i=1}^N W_i' = \prod_{i=1}^N (W_i + \Delta_{W_i})$ into the following form:

$$\prod_{i=1}^N W_i + \alpha \sum_{k=1}^K \sum_{i=1}^N \left( \prod_{j=1}^{i-1} W_j \right) G_i\varphi(\mathbf{y}_k)\phi(\mathbf{x}_k)^T \left( \prod_{j=i+1}^N W_j \right)^T \left( \prod_{j=i+1}^N W_j \right) \tag{1}$$

$$-\alpha\mathcal{L} \sum_{k=1}^K \sum_{i=1}^N \left( \prod_{j=1}^{i-1} W_j \right) \text{diag}(G_i\varphi(\mathbf{y}_k))^2 \left( \prod_{j=i}^N W_j \right) + O(\alpha^2), \tag{2}$$

This expansion allows us to derive the form of $\mathbf{z}^\star$ for input $\mathbf{x}^\star$:

$$\mathbf{z}^\star = \prod_{i=1}^N W_i\phi(\mathbf{x}^\star) + \alpha \sum_{k=1}^K \sum_{i=1}^N \left( \prod_{j=1}^{i-1} W_j \right) G_i\varphi(\mathbf{y}_k)\phi(\mathbf{x}_k)^T \left( \prod_{j=i+1}^N W_j \right)^T \left( \prod_{j=i+1}^N W_j \right) \phi(\mathbf{x}^\star) \tag{3}$$

$$-\alpha\mathcal{L} \sum_{k=1}^K \sum_{i=1}^N \left( \prod_{j=1}^{i-1} W_j \right) \text{diag}(G_i\varphi(\mathbf{y}_k))^2 \left( \prod_{j=i}^N W_j \right) \phi(\mathbf{x}^\star),$$

Note that appropriate choice of $W_i$ and $G_i$ allows us to simplify the form of $\bar{\mathbf{z}}^\star$ in Equation 3 into the following:

$$\mathbf{z}^\star = B_0\phi(\mathbf{x}^\star) + \alpha \sum_{k=1}^K \sum_{i=1}^N B_0(B_{i-1})^{-1} G_i\varphi(\mathbf{y}_k)\phi(\mathbf{x}_k)^T B_i^T B_i \phi(\mathbf{x}^\star) \tag{4}$$

$$-\alpha\mathcal{L} \sum_{k=1}^K \sum_{i=1}^N B_0(B_{i-1})^{-1} [\text{diag}(G_i\varphi(\mathbf{y}_k))]^2 B_{i-1}\phi(\mathbf{x}^\star) \tag{5}$$

where the $B_i = \left(\prod_{i+1}^{N} W_i\right)$ can be set to arbitrary invertible square matrices.

Now, our goal is to choose $B_i$, $G_i$, $\varphi$, and $\phi$ to ensure that the expression above contains a complete description of the values of $\{(\mathbf{x}, \mathbf{y})_k\}$ (up to permuting the order of the examples) and $\mathbf{x}^\star$. Since $f_{\text{out}}$ can approximate any function to arbitrary precision, $\hat{f}(\mathbf{x}^\star; \theta') = f_{\text{out}}(\mathbf{z}^\star)$ can approximate any function of $\{(\mathbf{x}, \mathbf{y})_k\}$ and $\mathbf{x}^\star$.

We set $\varphi(\mathbf{y}) = \text{discr}(\mathbf{y})$, yielding a one-hot $d$-dimensional vector indicating the value of $\mathbf{y}$ up to arbitrary precision. We let $\phi$ (recall $\phi$ is a universal function approximator) have the following form:

$$\phi(\mathbf{x}) \approx \begin{bmatrix} 0 \\ \text{discr}(\mathbf{x}) \\ \mathbf{0}_{J^2 d} \\ \text{discr}(\mathbf{x}) \end{bmatrix},$$

where $\text{discr}(\mathbf{x})$ is a one-hot $J$-dimensional vector indicating the value of $\mathbf{x}$ up to a discretization of arbitrary precision, and $\mathbf{0}_{J^2}$ is a zero vector of dimension $J^2$. Note that $\phi$ satisfies the requirement that all its outputs are nonnegative. We furthermore let $N = J^2$ and rewrite the layer index $i$ as a double index $(j, l)$ where $j$ and $l$ each range from 1 through $J$. For future reference let us denote the dimensionality of $\mathbf{y}$ as $d$. $B_{j,l}$ and $G_{j,l}$ are defined as follows:

$$B_{j,l} := \begin{bmatrix} 0 & \tilde{B}_{j,l} & 0 & 0 \\ 0 & 0_{J \times J} & 0 & 0 \\ 0 & 0 & 0_{J^2 d \times J^2 d} & 0 \\ 0 & 0 & 0 & I_{J \times J} \end{bmatrix} + \epsilon I \qquad G_{j,l} := \begin{bmatrix} 0_{1 \times d} \\ 0_{J \times d} \\ \tilde{G}_{j,l} \\ 0_{J \times d} \end{bmatrix} \qquad (6)$$

where $\tilde{B}_{j,l}$ is a $1 \times J$ matrix containing ones in the $j$ and $l$ positions and zeroes elsewhere, the $\epsilon I$ is included to ensure the invertibility of $B_{j,l}$, and $\tilde{G}_{j,l}$ maps $\varphi(\mathbf{y})$ to a vector consisting of a stack of $J^2$ $d$-dimensional vectors, all of which are zero except the vector in the slot corresponding to $(j, l)$, which is $\varphi(\mathbf{y})$. That is,

$$\tilde{G}_{j,l}\varphi(\mathbf{y}) := \begin{bmatrix} \mathbf{0}_d \\ \vdots \\ \mathbf{0}_d \\ \text{discr}(\mathbf{y}) \\ \mathbf{0}_d \\ \vdots \\ \mathbf{0}_d \end{bmatrix} \qquad (7)$$

with $\varphi(\mathbf{y})$ appearing in the $J * j + l$ position.

Now we observe that:

$$\phi(\mathbf{x})^T B_{jl}^T \approx \begin{cases} \mathbf{e}_j^T & \text{if } \text{discr}(\mathbf{x}) \in \{\mathbf{e}_j, \mathbf{e}_l\} \\ \mathbf{0} & \text{otherwise} \end{cases} \qquad B_{jl}\phi(\mathbf{x}^\star) \approx \begin{cases} \mathbf{e}_j & \text{if } \text{discr}(\mathbf{x}^\star) \in \{\mathbf{e}_j, \mathbf{e}_l\} \\ \mathbf{0} & \text{otherwise} \end{cases}$$

where the approximation in the equalities is due to the $\epsilon$ terms included to ensure invertibility.

As a result, we have:

$$\mathbf{z}^\star \approx B_0 \phi(\mathbf{x}^\star) + \alpha \sum_{k=1}^{K} \begin{bmatrix} 0 \\ \mathbf{0}_J \\ \tilde{\mathbf{z}}_k^\star \\ \mathbf{0}_J \end{bmatrix},$$

where $\tilde{\mathbf{z}}_k^\star \approx \begin{cases} v(\text{discr}(\mathbf{y}_k), \{j + J * l, l + J * j\}) & \text{if } \text{discr}(\mathbf{x}^\star) = e_j \neq e_l = \text{discr}(\mathbf{x}_k) \\ v(\text{discr}(\mathbf{y}_k), \{j + J * i | 1 \leq i \leq J\} \cup \{i + J * j | 1 \leq i \leq J\}) & \text{if } \text{discr}(\mathbf{x}^\star) = e_j = \text{discr}(\mathbf{x}_k) \end{cases}$

with $v(\mathbf{a}, A)$ defined as the $J^2 d$-dimensional vector consisting of $J^2$ stacked $d$-dimensional vectors, all of which are zero except those located in the slots specified by the set $A$, which are set to $\mathbf{a}$.

Now we claim that $\{(\mathbf{x}, \mathbf{y})_k\}$ and $\mathbf{x}^\star$ can be decoded with arbitrary accuracy from $\mathbf{z}^\star$. Indeed, note that $B_0 = \prod_{i=1}^N$ contains an identity matrix in its last $J$-dimensional block, meaning that $B_0 \phi(\mathbf{x}^\star)$, and hence $\mathbf{z}^\star$, contains an unaltered copy of $\text{discr}(\mathbf{x}^\star)$ in its last $J$ dimensions, from which $\mathbf{x}^\star$ can be decoded to arbitrary accuracy. Given the value of $\mathbf{x}^\star$ we may also subtract $B_0 \phi(\mathbf{x}^\star)$ from $\mathbf{z}^\star$ and multiply by $\frac{1}{\alpha}$ to obtain an unaltered version of $\sum_{k=1}^K \tilde{\mathbf{z}}_k^\star$. Next, we may decode $\sum_{k=1}^K \tilde{\mathbf{z}}_k^\star$ in the following fashion. First, we can infer whether, and if so how many, of the $\mathbf{x}_k$ have the same discretization as $\mathbf{x}^\star$ by checking if any of the $J$ $d$-dimensional vectors in slot $j + J * j$ is nonzero, and if so, what its value is. If slot $j + J * j$ has nonzero value $\mathbf{c}$, we subtract $\mathbf{c}$ from all slots with index $j + J * i$ and $i + J * j$ for any $i$. Given $\text{discr}(\mathbf{x}^\star) = e_l$ the resulting vector, which we may call $\tilde{\mathbf{z}}_k^{\star\star}$, This leaves us with a vector which in each slot $j + J * l$ and $l + J * j$ indicates (by summing the $d$ components of the slot) how many times an $\mathbf{x}$ has been observed with $\text{discr}(\mathbf{x}) = e_j$ and (by looking at the nonzero components in the slot) counts of how many times every possible $\text{discr}(\mathbf{y})$ value was observed to correspond with that $\text{discr}(\mathbf{x})$. Thus, the set $\{(\mathbf{x}, \mathbf{y})_k\}$ as well as $\mathbf{x}^\star$ can be decoded to arbitrary accuracy from $\mathbf{z}^\star$.

Since $f_{\text{out}}$ is a universal function approximator, we let $f_{\text{out}}(\mathbf{z}^\star)$ be the function that performs the decoding procedure above and then uses the inferred values of $\{(\mathbf{x}, \mathbf{y})_k\}$ and $\mathbf{x}^\star$ to approximate $f_{\text{target}}(\{(\mathbf{x}, \mathbf{y})_k\}, \mathbf{x}^\star)$ to arbitrary precision.

