# OpenReview forum: "Learning to Learn with Feedback and Local Plasticity"
_NeurIPS.cc/2019/Workshop/Neuro_AI — Real Neurons & Hidden Units @ NeurIPS 2019 Oral_

### Official Review · AnonReviewer3 · 2019-09-20
**Interesting paper, would be keen to learn more**

**Clarity:** 4

**Comment:**

Overall, a great submission. I have many questions (e.g., what does the learned feedback look like? What do the learned update rules look like? Why force local learning rules based on old findings in neuroscience? We are beginning to realize that it isn't all about Hebbian plasticity! See e.g.: https://science.sciencemag.org/content/357/6355/1033.abstract). But, the workshop is the perfect place to ask these questions. :)

**Category:**

Common question to both AI & Neuro

**Clarity Comment:**

Not perfect, but overall very well-written.

**Evaluation:**

4: Very good

**Importance:**

4: Very important

**Importance Comment:**

This submission presents a meta-learning approach to discovering local updates guided by feedback. The goal is to move towards more biologically plausible learning mechanisms. This is an important topic for linking neuroscience and AI, and the approach the authors take here is interesting/promising.

**Intersection:**

5: Outstanding

**Intersection Comment:**

Definitely at the intersection of neuroscience and AI!

**Rigor Comment:**

There was little in the way of technical details. Partly, that was a matter of space, but it was also partly a matter of choice (e.g. the description of the experiments could have been shortened to make a bit more room for math). Also, the proof only demonstrates the expressivity of the approach. One concern I would have is the question of training efficiency - is it more efficient than standard meta-learning techniques and are there ways to make it more efficient by loosening some of the constraints on feedback and plasticity rules? Regardless, it is hard to fully assess the technical rigour, but the basic concept seems sound and the experiments are reasonably convincing. The finding regarding the importance of feedback is particularly illuminating in my opinion.

**Technical Rigor:**

3: Convincing

---

### Official Review · AnonReviewer1 · 2019-09-25
**Nice work. Needs to be clearer about whether it's trying to solve credit assignment for general learning problems, or just online learning**

**Clarity:** 4

**Comment:**

Define the model more explicitly. And emphasize that this only solves credit assignment for certain types of learning problems (at the moment).

**Category:**

AI->Neuro

**Clarity Comment:**

The submission is pretty clear.

In understanding the model, it would be useful to more explicitly define the model. For instance, how is the b at line 63 related to the activation x_i and ReLU at lines 75 and 76?

**Evaluation:**

4: Very good

**Importance:**

4: Very important

**Importance Comment:**

This is nice work that addresses the credit assignment problem with a meta-learning approach. The motivation needs to be a bit clearer. Is the work trying to address the credit assignment problem in general, or just when applied to online learning tasks? Either way this is important work, with many interesting future directions.

**Intersection:**

4: High

**Intersection Comment:**

There are exiting directions in both AI and neuroscience this work could be take.

Seeing if these meta-learnt rules line up with previously characterized biological learning rules is particularly interesting.

**Rigor Comment:**

The model and implementation make sense as far as I can tell from this brief submission.

The theoretical results stated are nice to have.

Section 1 pitches the method as solving the credit assignment problem, citing problems with weight symmetry etc, that apply to many forms of learning. But the related work in Section 2 then goes on to talk about the efficiency of backprop for solving online learning and few-shot learning tasks. The efficiency of backprop should be mentioned in the intro if it is something this work is aiming to address.

While much human learning may be more naturally cast as online learning, not all of it is. There may be much interest in how we learn from so few samples in certain settings, but we also learn some relationships/tasks in a classical associationist manner which is well modeled by 'slow' gradient-descent like learning (e.g. Rescorla Wagner). The credit assignment problem exists in these cases also. So I think the present work needs to be repitched slightly as solving credit assignment in an online/few shot learning setting. Or discuss how it can be extended to more general learning problems.

**Technical Rigor:**

3: Convincing

---

### Official Review · AnonReviewer2 · 2019-09-25
**Meta-learning biologically plausible networks**

**Clarity:** 4

**Comment:**

This paper presents an interesting approach to improving biologically plausible learning in deep networks. A few aspects of the paper could be clarified, e.g. the baseline methods. Diagrams would also be helpful in clarifying the feedforward vs. feedback mechanisms. Again, I would want to see the additional analyses included in the final draft. This paper would be a useful addition to the workshop.

**Category:**

Common question to both AI & Neuro

**Clarity Comment:**

Much of the paper was clear in its description. One point of confusion is the distinction between gradient-based learning and gradient-based meta-learning. The authors claim that they compare with gradient-based meta-learning, however, their method also uses gradients to perform meta-learning. Clarifying these details/wording would help to clear up the confusion.

**Evaluation:**

5: Excellent

**Importance:**

5: Astounding importance

**Importance Comment:**

This paper presents some interesting results on meta-learning of weights in a more biologically plausible neural network. The results are fairly important, as they suggest that a proper initialization may be a key aspect of the success of biologically plausible learning rules.

**Intersection:**

4: High

**Intersection Comment:**

The paper touches on concepts in both neuroscience and machine learning, however, the paper ultimately seems more geared toward a machine learning audience. For instance, while the authors briefly speculate about alternative ways in which meta-learning could be implemented, they do not provide an in-depth discussion on its biological plausibility.

**Rigor Comment:**

Overall, the authors are rigorous in their evaluation. For the final draft, the authors should include their additional analyses on the feedback weights.

**Technical Rigor:**

4: Very convincing

---

### Decision · Program_Chairs · 2019-10-02

Accept (Oral)